# TABNET: ATTENTIVE INTERPRETABLE TABULAR LEARNING

## ABSTRACT

We propose a novel high-performance interpretable deep tabular data learning network, TabNet. TabNet utilizes a sequential attention mechanism that softly selects features to reason from at each decision step and then aggregates the processed information to make a final prediction decision. By explicitly selecting sparse features, TabNet learns very efficiently as the model capacity at each decision step is fully utilized for the most relevant features, resulting in a high performance model. This sparsity also enables more interpretable decision making through the visualization of feature selection masks. We demonstrate that TabNet outperforms other neural network and decision tree variants on a wide range of tabular data learning datasets, especially those are not saturated in performance, and yields interpretable feature attributions and insights into the global model behavior.

## 1 INTRODUCTION

Deep neural networks have been demonstrated to be very powerful in understanding images (He et al., 2015; Simonyan & Zisserman, 2014; Zagoruyko & Komodakis, 2016), text (Conneau et al., 2016; Devlin et al., 2018; Lai et al., 2015) and audio (van den Oord et al., 2016; Amodei et al., 2015; Chiu et al., 2018), yielding many important artificial intelligence use cases. For these data types, a major enabler of the rapid research and development progress is the availability of canonical neural network architectures to efficiently encode the raw data into meaningful representations. Integrated with simple decision-making layers, these canonical architectures yield high performance on new datasets and related tasks with small extra tuning effort. For example, consider the image understanding task – variants of convolutional layers with residual connections, e.g. the notable ResNet (He et al., 2015) architecture, can yield reasonably good performance on a new image dataset (e.g. in medical imaging) or a slightly different visual recognition problem (e.g. segmentation).

Our focus in this paper is tabular (structured) data. Tabular data is indeed the most common data type in the entire addressable artificial intelligent market (Chui et al., 2018). Yet, canonical neural network architectures for tabular data understanding have been under-explored. Instead, variants of ensemble decision trees still dominate data science competitions with tabular data (Kaggle, 2019b). A primary reason for the popularity of tree-based approaches is their representation power for decision manifolds with approximately hyperplane boundaries that are commonly observed for tabular data. In addition, decision tree-based approaches are easy to develop and fast to train. They are highly-interpretable in their basic form (e.g. by tracking decision nodes and edges) and various interpretability techniques have been shown to be effective for their ensemble form, e.g. (Lundberg et al., 2018). On the other hand, conventional neural network architectures based on stacked convolutional or multi-layer perceptrons, may not be the best fit for tabular data decision manifolds due to being vastly overparametrized – the lack of appropriate inductive bias often causes them to fail to find robust solutions for tabular decision manifolds (Goodfellow et al., 2016). We argue here that neural network architectures for tabular data should be redesigned to account for a 'decision-tree-like' mapping.

Given the aforementioned benefits and reasonable performances of tree-based methods, why is deep learning worth exploring for tabular data? One obvious motivation is pushing the performance albeit an increased computational cost, especially with more training data. In addition, introduction of a high-performance deep neural network architecture unlocks the benefits of gradient descent-based end-to-end deep learning for tabular data. For example, decision tree learning (even with gradient boosting) does not utilize back-propagation into their inputs to use an error signal to guide efficient learning of complex data types. On the other hand, with a deep neural network architecture, complex

data types like images can be integrated into tabular data efficiently. Another well-known challenge for tree-based methods is learning from streaming data. Most algorithms for tree learning need global statistical information to select split points and straightforward modifications such as (Ben-Haim & Tom-Tov, 2010) typically yield lower accuracy compared to learning from full data at once, yet deep neural networks show great potential for continual learning (Parisi et al., 2018). Lastly, deep learning models learn meaningful representations which enable new capabilities such as data-efficient domain adaptation (Goodfellow et al., 2016), generative modeling (e.g. using variational autoencoders or generative adversarial networks (Radford et al., 2015) or semi-supervised learning (Dai et al., 2017). As one example of these potential new capabilities, we demonstrate the potential of semi-supervised learning in the Appendix, showing the potential benefits of information extraction from unlabeled data which non-deep learning models are much weaker at.

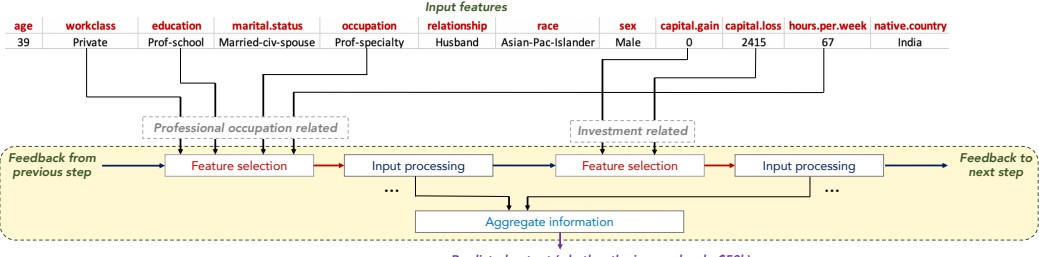

Figure 1: Depiction of TabNet's sparse feature selection for Adult Census Income prediction (Dua & Graff, 2017). TabNet employs multiple decision blocks that focus on processing a subset of input features for overall decision making. Two decision blocks shown as examples process the features that are professional occupation related and investment related in order to predict the income level.

In this paper we propose TabNet, a deep neural network architecture to make a significant leap forward towards the optimal model design for tabular data learning. Motivated by the key problems for tabular data discussed above, the design of TabNet has two goals that are often not considered jointly: state-of-the-art performance and interpretability. With decision tree motivations, TabNet brings sparsity-controlled soft feature selection, but individually for each instance. Unlike other instance-wise feature selection methods like (Chen et al., 2018) or (Yoon et al., 2019), TabNet employs a *single deep learning architecture with end-to-end learning*, to map the raw data to the final decision with *soft feature selection*. The key aspects and contributions of TabNet are:

1. In order to learn flexible representations and enable flexible integration into end-to-end learning, unlike most tabular data methods, *TabNet inputs raw tabular data without any feature preprocessing* and is *trained using conventional gradient descent-based optimization*.
2. To improve performance and interpretability, *TabNet utilizes a sequential attention mechanism to choose which features to reason from at each decision step*, as shown in Fig. 1. We design this feature selection to be instance-wise such that the model can decide which features to focus on separately for each input – e.g., for income classification capital gain may be a more important feature to focus on for a middle-aged individual. Explicit selection of sparse features enables interpretability as well as more efficient learning as the model parameters are fully utilized for the most salient features at the corresponding decision step.
3. Overall, our careful architecture design leads to two valuable properties for real world tabular learning problems: (1) *TabNet outperforms other tabular learning models* on various datasets for classification and regression problems from different domains, particularly those which are not saturated in performance; and (2) *TabNet enables two kinds of interpretability*: local interpretability that visualizes the importance of input features and how they are combined, and global interpretability that quantifies the amount of contribution of each input feature to the trained model.

## 2 RELATED WORK

**Feature selection:** Feature selection in machine learning broadly refers to the techniques for judicious selection of a subset of features that are useful to build a good predictor for a specified response variable. Commonly-used feature selection techniques such as forward feature selection and LASSO regularization (Guyon & Elisseeff, 2003) attribute importance to the features based on the entire training data set, and are referred as *global* methods. On the other hand, *instance-wise* feature selection refers to selection of the most important features, individually and differently for

each input. Instance-wise feature selection was studied in (Chen et al., 2018) by training an explainer model to maximize the mutual information between the selected features and the response variable, and in (Yoon et al., 2019) by using an actor-critic framework to mimic a baseline model while optimizing the feature selection. (Ma et al., 2018) uses partial variational autoencoder to dynamically decide which piece of information to acquire next sequentially, that can be adapted for instance-wise feature selection. Unlike these approaches, our proposed method employs *soft feature selection with controllable sparsity in end-to-end learning* – a single model jointly performs feature selection and output mapping, enabled by the specific design of the architecture. Thus, we demonstrate superior performance with very compact representations.

**Tree-based learning:** Tree-based models are the most common machine learning approaches for tabular data learning. The prominent strength of tree-based models is their efficacy in picking global features with the most statistical information gain (Grabczewski & Jankowski, 2005). To improve the performance of standard tree-based models by reducing the model variance, one common approach is ensembling. Among ensembling methods, random forests (Ho, 1998) use random subsets of data with randomly selected features to grow many trees. XGBoost (Chen & Guestrin, 2016) and LightGBM (Ke et al., 2017) are the two recent ensemble decision tree approaches that dominate most of the recent data science competitions. They are based on learning the structures of trees at first, and then updating the leaves with the streaming data.

**Integration of neural networks into decision trees:** One direction to address the limitations of decision trees is integration of neural networks. Representing decision trees with canonical neural network building blocks, as in (Humbird et al., 2018), yields redundancy in representation and inefficient learning. Soft (neural) decision trees (Wang et al., 2017; Kontschieder et al., 2015) are proposed with differentiable decision functions, instead of non-differentiable axis aligned splits to construct trees. Yet, abandoning axis-aligned splits loses the automatic feature selection ability, which is important for learning from tabular data. In (Yang et al., 2018), a soft binning function is proposed to simulate decision trees in neural networks, which needs to enumerate all possible decisions and is inefficient. In (Ke et al., 2019), a novel neural network architecture is proposed, with the motivations of explicitly leveraging expressive feature combinations and reducing model complexity. However, learning is based on transferring knowledge from a gradient boosted decision tree. Thus, it yields very limited performance improvement compared to it, and interpretability was not considered. In (Tanno et al., 2018), a deep learning framework is proposed based on adaptively growing the architecture from primitive blocks while representation learning into edges, routing functions and leaf nodes of a decision tree. Our proposed model TabNet differs from these methods as it embeds the soft feature selection ability into a sequential attention-based network architecture, with controllable sparsity.

**Attentive table-to-text models:** Table-to-text models extract textual information from tabular data. Recent works (Liu et al., 2017) (Bao et al., 2019) propose an architecture based on sequential mechanism for field-level attention. Despite the high-level similarities in the architecture, TabNet aims to perform the ultimate classification or regression task considering the entire input features, rather than mapping them to a different data type.

## 3 TabNet model

### 3.1 Principles

We initially consider the implementation of a decision tree-like output manifold using conventional neural network building blocks (Fig. 2). Individual feature selection is the key idea to obtain decision boundaries in hyperplane form, which can be generalized for linear combination of features where constituent coefficients determine the proportion of each feature in the decision boundary. We aim to generalize this type of tree-like functionality by:

- Utilizing sparse instance-wise feature selection, learned based on the training dataset.
- Constructing a sequential multi-step architecture, where each decision step can contribute to a portion of the decision that is based on the selected features.
- Improving the model capacity by non-linear processing of the selected features.
- Ensembling via higher feature dimension and more decision steps.

### 3.2 Overall architecture

Fig. 3 depicts the TabNet architecture. Tabular data inputs are comprised of numerical and categorical features. We use the raw numerical features and we consider mapping of categorical features with

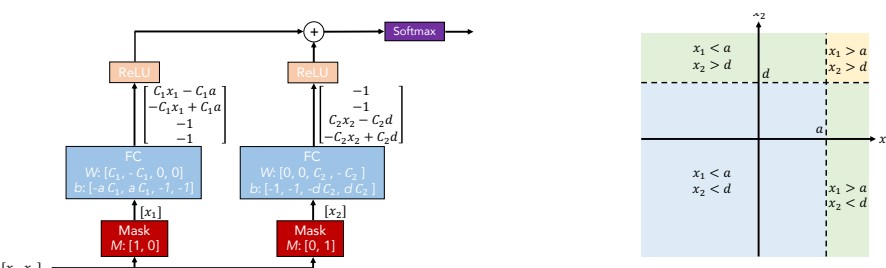

Figure 2: Illustration of decision tree-like classification using conventional neural network blocks and the corresponding two-dimensional manifold ($x_1$ and $x_2$ are the input dimensions, and $a$ and $d$ are constants). By employing multiplicative sparse masks to inputs, the relevant features are selected. The selected features are linearly transformed and after a bias addition (to represent boundaries), ReLU function performs region selection by zeroing the regions that are on the negative side of the boundary. Aggregation of multiple regions is based on the addition operation. As $C_1$ and $C_2$ get larger, the decision boundary gets sharper due to the softmax.

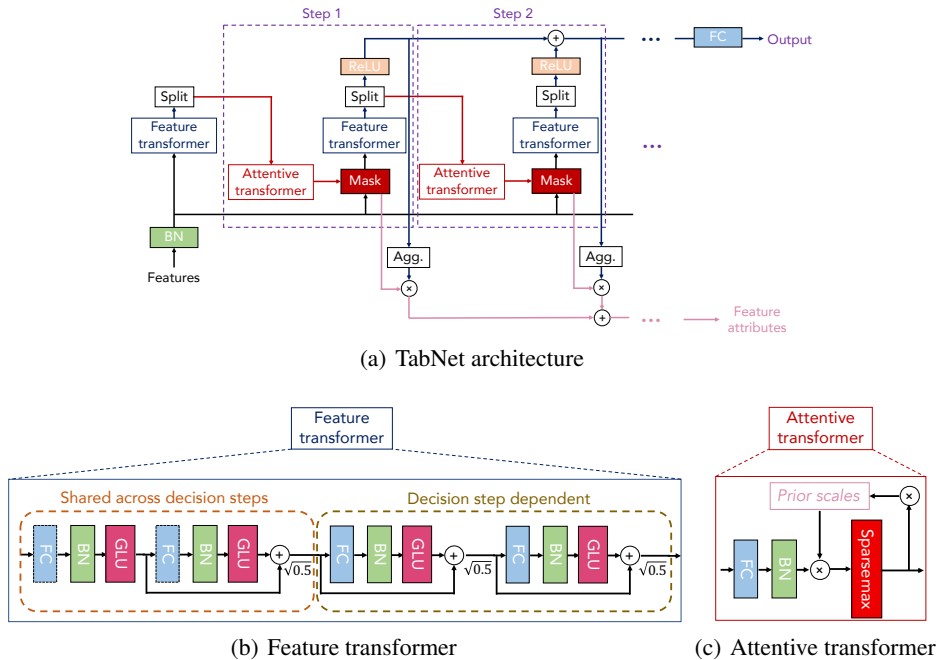

Figure 3: (a) TabNet architecture, composed of feature transformer, attentive transformer and feature masking at each decision step. Split block divides the processed representation into two, to be used at the attentive transformer of the subsequent step and to be used towards construction of the overall output. At each decision step, its feature selection mask can provide insights about its functionality, and the masks can be aggregated (using the Agg. block) ultimately to obtain global feature important attribution behavior. (b) A feature transformer block example – 4-layer network is shown, where 2 of them are shared across all decision steps and 2 of them are decision step-dependent. Each layer is composed of a fully-connected layer, batch normalization and GLU nonlinearity. (c) An attentive transformer block example – a single layer mapping is modulated with a prior scale information, which aggregates how much each feature has been used before the current decision step. Normalization of the coefficients is employed using sparsemax (Martins & Astudillo, 2016) for sparse selection of the most salient features at each decision step.

trainable embeddings[1]. We do not consider any global normalization of features, but merely apply batch normalization. We pass the same $D$-dimensional features $\mathbf{f} \in \Re^{B \times D}$ to each decision step, where $B$ is the batch size. TabNet is based on sequential multi-step processing, with $N_{steps}$ decision steps. The $i^{th}$ step inputs the processed information from the $(i-1)^{th}$ step to decide which features to use and outputs the processed feature representation to be aggregated into the overall decision. The idea of top-down attention in sequential form is inspired from its applications in processing visual and language data such as for visual question answering (Hudson & Manning, 2018) or in reinforcement learning (Mott et al., 2019) while searching for a small subset of relevant information in high dimensional input. Ablation studies in Appendix focus on the impact of various TabNet design choices, explained next. Overall, the performance is not too sensetive to most hyperparameters, and guidelines on selection of the important hyperparameters are also provided in Appendix.

**Feature selection:** We employ a learnable sparse mask $\mathbf{M[i]} \in \Re^{B \times D}$ for soft selection of the salient features. Through sparse selection of the most salient features, the learning capacity of a decision step is not wasted on irrelevant features, and thus the model becomes more parameter efficient. The masking is in multiplicative form, $\mathbf{M[i]} \cdot \mathbf{f}$. We use an attentive transformer (see Fig. 3) to obtain the masks using the processed features from the preceding step, $\mathbf{a[i-1]}$:

$$\mathbf{M[i]} = \mathrm{sparsemax}(\mathbf{P[i-1]} \cdot \mathrm{h}_i(\mathbf{a[i-1]})). \tag{1}$$

Sparsemax normalization (Martins & Astudillo, 2016) encourages sparsity by mapping the Euclidean projection onto the probabilistic simplex, which is observed to be superior in performance and aligned with the goal of sparse feature selection for most real-world datasets. Note that Eq. 1 has the normalization property, $\sum_{j=1}^{D} \mathbf{M[i]_{b,j}} = 1$. $\mathrm{h}_i$ is a trainable function, shown in Fig. 3 using a fully-connected layer, followed by batch normalization. $\mathbf{P[i]}$ is the prior scale term, denoting how much a particular feature has been used previously:

$$\mathbf{P[i]} = \prod_{j=1}^{i} (\gamma - \mathbf{M[j]}), \tag{2}$$

where $\gamma$ is a relaxation parameter – when $\gamma = 1$, a feature is enforced to be used only at one decision step and as $\gamma$ increases, more flexibility is provided to use a feature at multiple decision steps. $\mathbf{P[0]}$ is initialized as all ones. To further control the sparsity of the selected features, we propose sparsity regularization in the form of entropy (Grandvalet & Bengio, 2004):

$$L_{sparse} = -\frac{1}{N_{steps} \cdot B} \sum_{i=1}^{N_{steps}} \sum_{b=1}^{B} \sum_{j=1}^{D} \mathbf{M_{b,j}[i]} \log(\mathbf{M_{b,j}[i]} + \epsilon), \tag{3}$$

where $\epsilon$ is a small number for numerical stability. We add the sparsity regularization to the overall loss, with a coefficient $\lambda_{sparse}$. Sparsity may provide a favorable inductive bias for convergence to higher accuracy for some datasets where most of the input features are redundant.

**Feature processing:** We process the filtered features using a feature transformer (see Fig. 3) to obtain the features that are split for the decision step output and information for the subsequent step:

$$[\mathbf{d[i]}, \mathbf{a[i]}] = \mathrm{f}_i(\mathbf{M[i]} \cdot \mathbf{f}), \tag{4}$$

where $\mathbf{d[i]} \in \Re^{B \times N_d}$ and $\mathbf{a[i]} \in \Re^{B \times N_a}$. For parameter-efficient learning and efficient convergence with the high model capacity, a feature transformer should comprise layers that are shared across all decision steps (as the same features are input across different decision steps), as well as decision step-dependent layers. In Fig. 3, we show the implementation of a block as concatenation of two shared layers and two decision step-dependent layers. Each fully-connected layer is followed by batch normalization and gated linear unit (GLU) nonlinearity (Dauphin et al., 2016) [2], eventually connected to a normalized residual connection with normalization. Normalization with $\sqrt{0.5}$ helps to stabilize learning by ensuring that the variance throughout the network does not change dramatically, as discussed in (Gehring et al., 2017). For faster training, we aim for very large batch sizes in practice. To improve performance with large batch sizes, all batch normalization operations, except the one applied to the input features, are implemented in ghost batch normalization (Hoffer et al., 2017) form, with a virtual batch size $B_V$ and momentum $m_B$. For the input features, we observe the benefit of

---

[1]For example, if there are three possible categories A, B and C for a particular features, they can be learned to be mapped to scalars 0.4, 0.1, and -0.2 respectively, along with the training of the model.

[2]In GLU, first a linear mapping is applied to the intermediate representation and the dimensionality is doubled, and then second half of the output is used to determine nonlinear processing on the first half.

low-variance averaging and hence avoid ghost batch normalization. After the last layer, we split the processed representation into $\mathbf{d[i]}$ and $\mathbf{a[i]}$. Inspired by decision-tree like aggregation as in Fig. 2, we construct the overall decision embedding as:

$$\mathbf{d_{out}} = \sum_{i=1}^{N_{steps}} \text{ReLU}(\mathbf{d[i]}). \tag{5}$$

Finally, we apply a linear mapping $\mathbf{W_{final}d_{out}}$, for the final decision. When discrete outputs are required, we additionally employ a softmax function during training (and argmax during inference).

## 4 EXPERIMENTS

We evaluate the performance of TabNet in wide range of problems, that contain regression or classification tasks. We specifically focus on tabular datasets with published benchmarks, based on notable tree-based and neural network-based approaches.

For all datasets, categorical inputs are mapped to a single-dimensional trainable scalar with a learnable embedding[3] and numerical columns are input without and preprocessing.[4] We use standard classification (softmax cross entropy) and regression (mean squared error) loss functions and we train until convergence. Hyperparameters of the TabNet models are optimized on a validation set and listed in Appendix. TabNet performance is not very sensitive to most hyperparameters as shown with ablation studies in Appendix. In all of the experiments where we cite results from other papers, we use the same training, validation and testing data split with the original work. Adam optimization algorithm (Kingma & Ba, 2014) and Glorot uniform initialization are used for training of all models.

### 4.1 PERFORMANCE

**Comparison to methods that integrate explicit feature selection.** For this comparison, we consider the 6 synthetic tabular datasets from (Chen et al., 2018). As the datasets are small (10k training samples), efficient feature selection is crucial for high performance in this task. The synthetic datasets are constructed in such a way that only a subset of the features determine the output. For Syn1, Syn2 and Syn3 datasets, the 'salient' features are the same for all instances, so that an accurate global feature selection mechanism should be optimal. E.g., the ground truth output of the Syn2 dataset only depends on features $X_3$-$X_6$. For Syn4, Syn5 and Syn6 datasets, the salient features are instance dependent. E.g., for Syn4 dataset, $X_{11}$ is the indicator, and the ground truth output depends on either $X_1$-$X_2$ or $X_3$-$X_6$ depending on the value of $X_{11}$. This instance dependence makes global feature selection suboptimal, as the globally-salient features would be redundant for some instances.

Table 1: TabNet achieves high performance with small number of network parameters. Mean and std. of test area under the receiving operating characteristic curve (AUC) on 6 synthetic datasets from (Chen et al., 2018), for TabNet vs. other feature selection-based neural network models: No sel.: using all input features without any feature selection, Global: using only globally-salient features, Tree refers to Tree Ensembles (Geurts et al., 2006), LASSO: LASSO-regularized model, L2X (Chen et al., 2018) and INVASE (Yoon et al., 2019) are instance-wise feature selection frameworks. Bold numbers are the best method for each dataset.

| Model | Syn1 | Syn2 | Syn3 | Syn4 | Syn5 | Syn6 |
|---|---|---|---|---|---|---|
| No sel. | .578 ± .004 | .789 ± .003 | .854 ± .004 | .558 ± .021 | .662 ± .013 | .692 ± .015 |
| Tree | .574 ± .101 | .872 ± .003 | .899 ± .001 | .684 ± .017 | .741 ± .004 | .771 ± .031 |
| Lasso | .498 ± .006 | .555 ± .061 | .886 ± .003 | .512 ± .031 | .691 ± .024 | .727 ± .025 |
| L2X | .498 ± .005 | .823 ± .029 | .862 ± .009 | .678 ± .024 | .709 ± .008 | .827 ± .017 |
| INVASE | **.690 ± .006** | .877 ± .003 | **.902 ± .003** | **.787 ± .004** | .784 ± .005 | .877 ± .003 |
| Global | .686 ± .005 | .873 ± .003 | .900 ± .003 | .774 ± .006 | .784 ± .005 | .858 ± .004 |
| *TabNet* | .682 ± .005 | **.892 ± .004** | .897 ± .003 | .776 ± .017 | **.789 ± .009** | **.878 ± .004** |

Table 1 shows the performance of TabNet vs. other techniques, including no selection, using only globally-salient features, Tree Ensembles (Geurts et al., 2006), LASSO regularization, L2X (Chen

---

[3]In some problems, higher dimensional embedding mapping may slightly improve the performance, but interpretation of individual embedding dimensions may become challenging.

[4]Specially-designed feature engineering, e.g. logarithmic transformation of variables highly-skewed distributions, may further improve the results but we leave it out of the scope of this paper.

et al., 2018) and INVASE (Yoon et al., 2019). We observe that TabNet outperforms all other methods and is on par with INVASE. For Syn1, Syn2 and Syn3 datasets, we observe that the TabNet performance is very close to global feature selection. For Syn4, Syn5 and Syn6 datasets, we observe that TabNet improves global feature selection, which would contain redundant features. (Feature selection is visualized in Sec. 4.2.) All other methods utilize a predictive model with 43k parameters, and the total number of trainable parameters is 101k for INVASE due to the two other networks in the actor-critic framework. On the other hand, TabNet is a single neural network architecture, and the total number of parameters is 26k for Syn1-Syn3 datasets and 31k for Syn4-Syn6 datasets. This compact end-to-end representation is one of TabNet's valuable properties.

**Comparison to models that do not employ explicit feature selection.** We compare TabNet to high-performance tabular data learning models that are demonstrated on the following problems:

- *Forest cover type* (Dua & Graff, 2017): Classification of forest cover type from cartographic variables.
- *Poker hand* (Dua & Graff, 2017): Classification of the poker hand from the raw input features of suit and rank attributes of the cards.
- *Sarcos robotics arm inverse dynamics* (Vijayakumar & Schaal, 2000): Regression for inverse dynamics of seven degrees-of-freedom of an anthropomorphic robot arm.
- *Higgs boson* (Baldi et al., 2014): Distinguishing between a signal process which produces Higgs bosons and a background process which does not.

Table 2: Performance for forest cover type dataset. The performance of the comparison models* are from (Mitchell et al., 2018). AutoInt models pairwise feature interactions with an attention-base deep neural network (Song et al., 2018). AutoML Tables (denoted as **) is an automated machine learning development tool based on ensemble of models including linear feed-forward deep neural network, gradient boosted decision tree, AdaNet (Cortes et al., 2016) and ensembles (AutoML, 2019). For AutoML Tables (**), the amount of node hours reflects the measure of the count of searched models for the ensemble and their complexity.[6] A single TabNet model, without fine-grained hyperparameter search, can outperform the accuracy of ensemble models with very thorough hyperparameter search.

| Model | Test accuracy (%) |
|---|---|
| XGBoost* | 89.34* |
| LightGBM* | 89.28* |
| CatBoost* | 85.14* |
| AutoInt | 90.24* |
| AutoML Tables (2 node hours)** | 94.56** |
| AutoML Tables (5 node hours)** | 94.95** |
| AutoML Tables (10 node hours)** | 96.67** |
| AutoML Tables (30 node hours)** | 96.93** |
| *TabNet* | **96.99** |

Table 3: Performance for poker hand induction dataset. The input-output relationship is deterministic and hand-crafted rules implemented with several lines of code can get 100% accuracy. Yet, neural networks and decision tree models severely suffer from the imbalanced data and cannot learn the required sorting and ranking operations with the raw input features. The results for comparison models* are from (Yang et al., 2018).

| Model | Test accuracy (%) |
|---|---|
| Decision tree* | 50.0* |
| Multi layer perceptron* | 50.0* |
| Deep neural decision tree* | 65.1* |
| XGBoost | 71.1 |
| LightGBM | 70.0 |
| CatBoost | 66.6 |
| *TabNet* | **99.3** |
| Rule-based | 100.0 |

Tables 2-5 show the performance comparisons. We observe that TabNet outperforms multi-layer perceptrons and the variants of ensemble decision trees on all four datasets. TabNet allocates the learning capacity to salient features, and it yields a more compact model in terms of the number of parameters. When the model size is constrained, we observe the superior performance of TabNet even

Table 4: Performance for Sarcos robotics arm inverse dynamics dataset. Three TabNet models of different sizes are considered (denoted with -S, -M and -L). The performance of the comparison models* are from (Tanno et al., 2018).

| Model | Test MSE | Number of parameters |
|---|---|---|
| Random forest* | 2.39* | 16.7K |
| Stochastic decision tree* | 2.11* | 28K |
| Multi layer perceptron* | 2.13* | 0.14M |
| Adaptive neural tree ensemble* | 1.23* | 0.60M |
| Gradient boosted tree* | 1.44* | 0.99M |
| *TabNet-S* | **1.25** | **6.3K** |
| *TabNet-M* | **0.28** | **0.59M** |
| *TabNet-L* | **0.14** | **1.75M** |

Table 5: Performance on Higgs boson dataset. Two TabNet models of different sizes are considered (denoted with -S and -M). The performance of the comparison models* are from (Mocanu et al., 2018). Sparse evolutionary training applies non-structured sparsity integrated into training, yielding low number of parameters. With its compact representation, TabNet, (without any further pruning or extra non-structured sparsity), yields almost similar performance with sparse evolutionary training for the same number of parameters. Gradient boosted tree models are implemented using (Tensorflow, 2019), see Appendix for details.

| Model | Test accuracy (%) | Number of parameters |
|---|---|---|
| Sparse evolutionary trained multi layer perceptron* | **78.47*** | **81K** |
| Gradient boosted tree-S | 74.22 | 0.12M |
| Gradient boosted tree-M | 75.97 | 0.69M |
| Multi-layer perceptron* | 78.44* | 2.04M |
| Gradient boosted tree-L | 76.98 | 6.96M |
| *TabNet-S* | 78.25 | 81K |
| *TabNet-M* | **78.84** | **0.66M** |

compared to the decision tree variants. The performance is only slightly worse than the evolutionary sparsification algorithms (Mocanu et al., 2018). Yet, the sparsity learned in TabNet is structured unlike the alternative approaches – i.e. it does not degrade the operational intensity of the model (Wen et al., 2016) and can efficiently utilize modern multi-core processors. Also note that we do not consider any matrix sparsification techniques such as adaptive pruning (Narang et al., 2017) which could further improve the parameter-efficiency.

## 4.2 INTERPRETABILITY

The feature selection masks in TabNet can be used to build insights on selected features at each step. Such capability would not be possible for conventional neural networks with fully-connected layers, as each subsequent layer hidden units would jointly process all features without sparsity-controlled selection mechanism. For feature selection masks, if $\mathbf{M_{b,j}[i]} = 0$, then $j^{th}$ feature of the $b^{th}$ sample should have 0 contribution to the overall decision. If $f_i$ were a linear function, the coefficient $\mathbf{M_{b,j}[i]}$ would correspond to the feature importance of $\mathbf{f_{b,j}}$. Although each decision step employs non-linear processing, their outputs are combined later in a linear way. Our goal is to quantify an aggregate feature importance beyond analysis of each step as well. Combination of the masks at different decision steps require a coefficient that can weigh the relative importance of each step in the decision. We use $\eta_\mathbf{b}[\mathbf{i}] = \sum_{c=1}^{N_d} \text{ReLU}(\mathbf{d_{b,c}[i]})$ to denote the aggregate decision contribution at $i^{th}$ decision step for the $b^{th}$ sample. Intuitively, $\mathbf{d_{b,c}[i]} < 0$, then all features at $i^{th}$ decision step should have 0 contribution to the overall decision and as its value increases, it plays a higher role in the overall linear combination given in Eq. 5. Scaling the decision mask at each decision step with $\eta_\mathbf{b}[\mathbf{i}]$, we propose the aggregate feature importance mask as:

$$\mathbf{M_{agg-b,j}} = \sum_{i=1}^{N_{steps}} \eta_\mathbf{b}[\mathbf{i}]\mathbf{M_{b,j}[i]} \Big/ \sum_{j=1}^{D} \sum_{i=1}^{N_{steps}} \eta_\mathbf{b}[\mathbf{i}]\mathbf{M_{b,j}[i]}. \qquad (6)$$

Normalization is used to ensure $\sum_{j=1}^{D} \mathbf{M_{agg-b,j}} = 1$ for each sample.

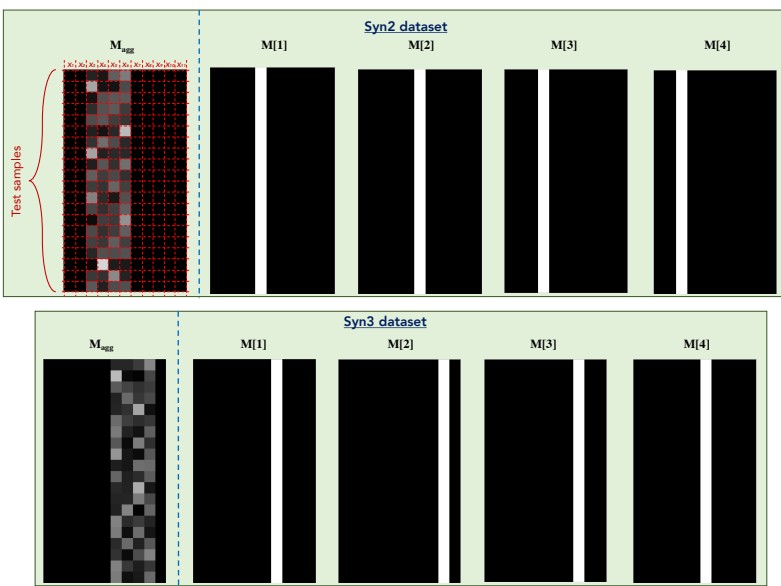

Figure 4: Feature importance masks $\mathbf{M}[\mathbf{i}]$ (that indicate which features are selected at $i^{th}$ step) and the aggregate feature importance mask $\mathbf{M_{agg}}$ showing the global instance-wise feature selection for Syn2 and Syn3 datasets from (Chen et al., 2018). Brighter colors show a higher value. E.g. for Syn2 dataset, only four features ($X_3$-$X_6$) are used.

.

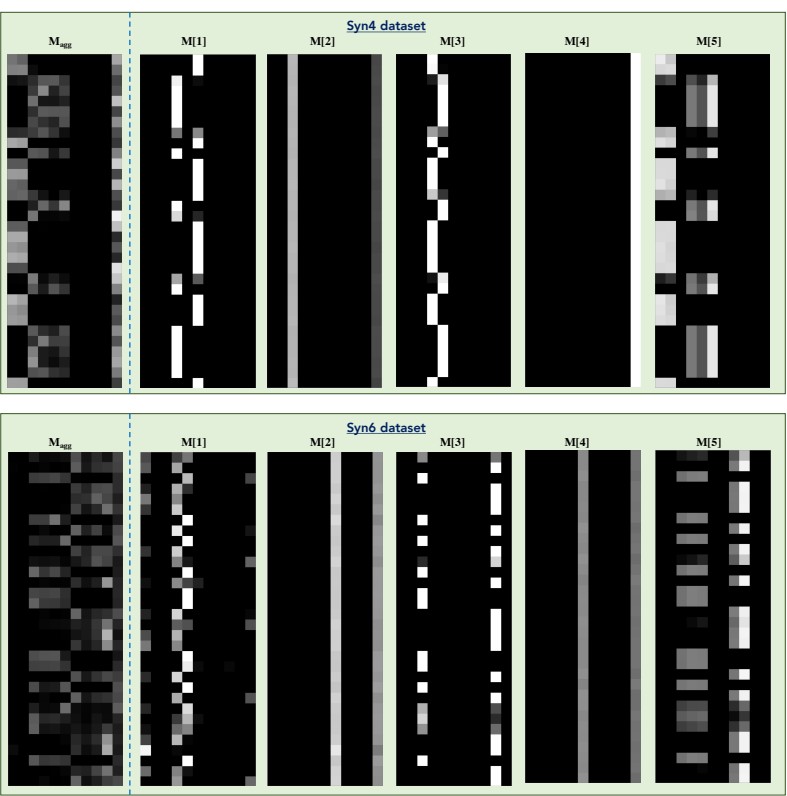

Figure 5: Feature importance masks $\mathbf{M}[\mathbf{i}]$ (that indicate which features are selected at $i^{th}$ step) and the aggregate feature importance mask $\mathbf{M_{agg}}$ showing the global instance-wise feature selection for Syn4 and Syn6 datasets from (Chen et al., 2018). Brighter colors show a higher value. E.g. for Syn4 dataset, the chosen features depend on the value of $X_{11}$.

.

Figs. 4 and 5 show the aggregate feature importance masks for the synthetic datasets discussed in Sec. 4.1 (for better illustration here, unlike Sec. 4.1, the models are trained with 10M training samples rather than 10K as we obtain sharper feature selection masks). The ground truth output of the Syn2 dataset only depends on features $X_3$-$X_6$, and the ground truth output of the Syn3 dataset only depends on features $X_7$-$X_{10}$. We observe that the aggregate masks are almost all-zero for irrelevant features and they merely focus on relevant ones. For Syn4 dataset, $X_{11}$ is the indicator, and the ground truth output depends on either $X_1$-$X_2$ or $X_3$-$X_6$ depending on the value of $X_{11}$. For Syn6 dataset, $X_{11}$ is the indicator, and the ground truth output depends on either $X_3$-$X_6$ or $X_7$-$X_{10}$ depending on the value of $X_{11}$. For both, TabNet yields accurate instance-wise feature selection – it uses majority of the weights in two masks to focus on $X_{11}$, and assigns almost all-zero weights to irrelevant features (the ones other than one of the two feature groups based on the value of $X_{11}$).

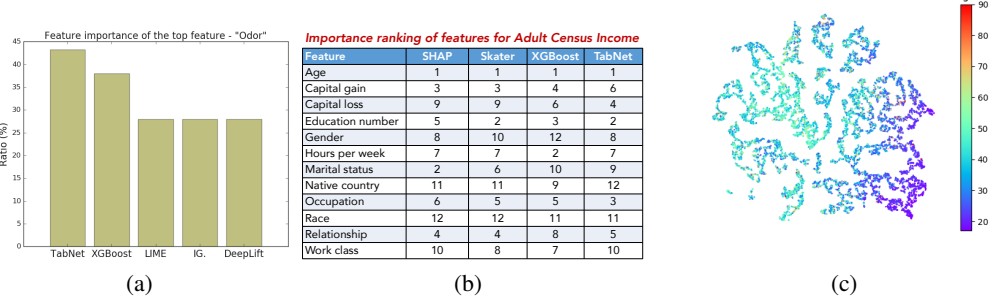

(a)  (b)  (c)

Figure 6: (a) Comparison to previous work for the ratio of the feature importance of "Odor" feature to all the features of the top feature for the mushroom edibility prediction (Dua & Graff, 2017) (task: classify whether a mushroom is edible or poisonous). With "Odor" feature only, $> 98.5\%$ test accuracy can be obtained, so a high feature importance is expected to be assigned to it, as observed with TabNet. (b) Comparison to previous work for importance ranking of features in the Adult Census Income dataset (Dua & Graff, 2017) (task: distinguish whether a person's income is above $50,000). (c) Impact of the most important feature on the decision manifold. T-SNE of the decision manifold for Adult Census Income test samples and the impact of the most dominant feature 'Age'.

Fig. 6(a) shows the feature importance score of the top feature obtained with TabNet vs. other explainability techniques from (Ibrahim et al., 2019) for mushroom edibility prediction. Mushroom edibility is a simple pattern recognition problem - TabNet achieves 100% test accuracy. It is indeed known (Dua & Graff, 2017) that with "Odor" feature only, a model can get $> 98.5\%$ test accuracy (Dua & Graff, 2017), so a high feature importance is expected for it, as observed with TabNet. Fig. 6(b) shows the importance ranking of features for TabNet vs. other explainability techniques from (Lundberg et al., 2018) (Nbviewer, 2019) for Adult Census Income prediction. TabNet achieves 85.7% test accuracy for this problem. We observe the commonality of the most important features ("Age", "Capital gain/loss", "Education number", "Relationship") and the least important features ("Native country", "Race", "Gender", "Work class"). For the same problem, Fig. 6(c) shows the impact of the most important feature on the output decision by visualizing the T-SNE of the decision manifold. A clear separation between age groups is observed, underlining the importance of the "Age" feature, as suggested by its high value in the aggregate feature importance mask of TabNet.

## 5 CONCLUSIONS

We propose TabNet, a novel deep learning architecture for tabular learning. TabNet utilizes a sequential attention mechanism to choose a subset of semantically-meaningful features to process at each decision step. The selected features are processed to the representation, that contributes to the overall decision output and sends information to the next decision step. Instance-wise feature selection enables efficient learning as the model capacity is fully used for the most salient features, and also yields more interpretable decision making via visualization of selection masks. We demonstrate that TabNet outperforms previous work across tabular datasets from different domains.

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

## A    SIMPLIFIED DIAGRAM FOR TABNET FEEDFORWARD PASS

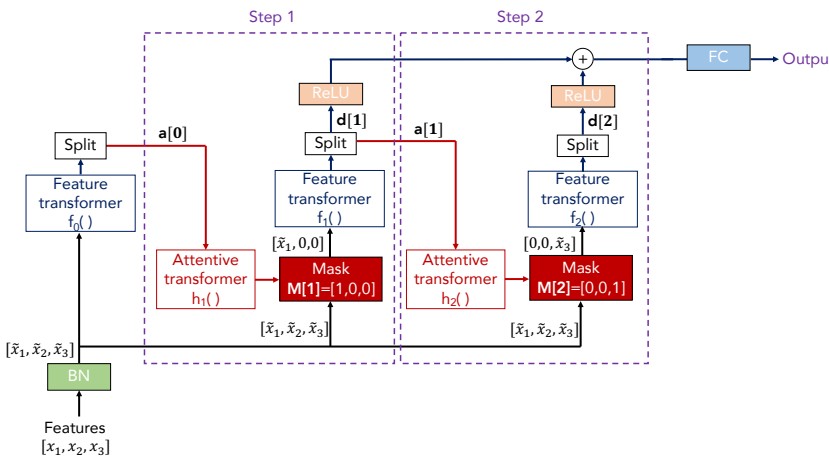

Figure 7: Simplified diagram for TabNet feedforward pass for an input with 3 features, assuming $N_{steps} = 2$. At the first step, the model selects only the first feature, and applies feature processing on it. At the second step, the model selects the last feature, and applies the feature processing on it. Lastly, the two outputs are combined for the final decision.

## B    ADDITIONAL RESULTS

### B.1    RETAIL DATASET WITH TIME COMPONENT

In this section, we show additional results on a real-world tabular data learning problem - Rossmann store sales forecasting (Kaggle, 2019a). This dataset has time-dependent features. Time information is input as day, month, and year columns. We observe that TabNet outperforms alternative methods that are commonly used for such problems.

Table 6: Performance for Rossmann store sales dataset (Kaggle, 2019a). We use the exactly same preprocessing and data split with (Catboost, 2019) - data from 2014 is used for training and validation, whereas 2015 is used for testing. The performance of the comparison models* are from (Catboost, 2019).

| Model | Test MSE |
|---|---|
| XGBoost* | 490.83* |
| LightGBM* | 504.76* |
| CatBoost* | 489.75* |
| *TabNet* | **485.12** |

### B.2    KDD DATASETS

Table 7: Performance on three KDD datasets on Customer Relationship Management: Appetency, Churn and Upselling. We apply the similar preprocessing and data partitioning as (Prokhorenkova et al., 2018). The performance of the comparison models* are from (Prokhorenkova et al., 2018).

| Model | Appetency test accuracy (%) | Churn test accuracy (%) | Upselling test accuracy (%) |
|---|---|---|---|
| XGBoost* | **98.2*** | 92.7* | **95.1*** |
| CatBoost* | **98.2*** | **92.8*** | **95.1*** |
| *TabNet* | **98.2** | 92.7 | 95.0 |

We experiment TabNet on four KDD datasets: the three Customer Relationship Management and Census Income. These datasets show saturated behavior in achievable performance (even simple

Table 8: Performance for KDD Census Income (Dua & Graff, 2017). The task is income prediction from demographic and employment related variables. The performance of the comparison models[*] are from (Oza, 2005).

| Model | Test accuracy (%) |
|---|---|
| XGBoost | **95.76** |
| CatBoost | 95.72 |
| Multi-layer perceptron[*] | 95.19 |
| Boosting, Multi-layer perceptron[*] | 94.86 |
| Bagging, Multi-layer perceptron[*] | 95.33 |
| *TabNet* | 95.49 |

models yield similar results). For these cases, TabNet shows very similar (or slightly worse) performance than XGBoost and CatBoost, that are known to be very robust as they contain high amount of ensembles.

### B.3 LOAN DELINQUENCY PREDICTION

Table 9: Performance for loan delinquency prediction on a proprietary dataset, constructed from (Mac, 2019). The task is to classify loan delinquency status (among four categories), from many input features including personal information and financial status. The training dataset consists of 93k samples. The dataset is highly imbalanced as the delinquency situation is observed rarely.

| Model | Test mean per class accuracy |
|---|---|
| XGBoost | 0.55 |
| H2OAutoML (with 15 models) | 0.60 |
| Multi-layer perceptron | 0.46 |
| *TabNet* | **0.86** |

We consider TabNet for a real-world problem in financial services industry: loan delinquency prediction. On a proprietary dataset, we demonstrate strong outperformance of TabNet, especially finding rare delinquency cases without any special techniques on anomaly detection.

## C EXPERIMENT HYPERPARAMETERS

For all datasets, we start hyperparameter tuning with a pre-defined value space. $N_d$ and $N_a$ are chosen from $\{8, 16, 24, 32, 64, 128\}$, $N_{steps}$ is chosen from $\{3, 4, 5, 6, 7, 8, 9, 10\}$, $\gamma$ is chosen from $\{1.0, 1.2, 1.5, 2.0\}$, $\lambda_{sparse}$ is chosen from $\{0, 0.000001, 0.0001, 0.001, 0.01, 0.1\}$, $B$ is chosen from $\{256, 512, 1024, 2048, 4096, 8192, 16384, 32768\}$, $B_V$ is chosen from $\{256, 512, 1024, 2048, 4096\}$ and $m_B$ is chosen from $\{0.6, 0.7, 0.8, 0.9, 0.95, 0.98\}$. If the model size is not under the desired cutoff (e.g. for Table 5 comparisons), we decrease the value to satisfy the model size constraint.

### C.1 SYNTHETIC DATASETS

All TabNet models use $N_d = N_a = 16$, $B = 3000$, $B_V = 100$, $m_B = 0.7$. For Syn1 we use $\lambda_{sparse} = 0.02$, $N_{steps} = 4$ and $\gamma = 2.0$; for Syn2 and Syn3 we use $\lambda_{sparse} = 0.01$, $N_{steps} = 4$ and $\gamma = 2.0$; and for Syn4, Syn5 and Syn6 we use $\lambda_{sparse} = 0.005$, $N_{steps} = 5$ and $\gamma = 1.5$. Each feature transformer block uses two shared and two decision step-dependent fully-connected layer, ghost batch normalization and GLU blocks. All models use Adam optimization a learning rate of 0.02 (decayed 0.7 every 200 iterations with an exponential decay) for 4k iterations.

For visualizations in Section 4.2, we also train TabNet models with datasets of size 10M samples. For this case, we choose $N_d = N_a = 32$, $\lambda_{sparse} = 0.001$, $B = 10000$, $B_V = 100$, $m_B = 0.9$. Adam optimization is used with a learning rate of 0.02 (decayed 0.9 every 2k iterations with an exponential decay) for 15k iterations. For Syn2 and Syn3, $N_{steps} = 4$ and $\gamma = 2$. For Syn4 and Syn6, $N_{steps} = 5$ and $\gamma = 1.5$.

## C.2  Forest cover type dataset

We use the exact same partitioning of the train, evaluation and test datasets with (Mitchell et al., 2018) for a fair comparison.

TabNet model uses $N_d = N_a = 64$, $\lambda_{sparse} = 0.0001$, $B = 16384$, $B_V = 512$, $m_B = 0.7$, $N_{steps} = 5$ and $\gamma = 1.5$. Each feature transformer block uses two shared and two decision step-dependent fully-connected layer, ghost batch normalization and GLU blocks. Adam optimization is used with a learning rate of 0.02 (decayed 0.95 every 0.5k iterations with an exponential decay) for 130k iterations.

## C.3  Poker hands dataset

TabNet uses $N_d = 24$, $N_a = 8$, $\lambda_{sparse} = 0.001$, $B = 4096$, $B_V = 256$, $m_B = 0.8$, $N_{steps} = 4$ and $\gamma = 1.5$. Each feature transformer block uses two shared and two decision step-dependent fully-connected layer, ghost batch normalization and GLU blocks. Adam optimization is used with a learning rate of 0.02 (decayed 0.9 every 10k iterations with an exponential decay) for 71k iterations.

## C.4  Sarcos dataset

TabNet-S model uses $N_d = N_a = 8$, $\lambda_{sparse} = 0.0001$, $B = 4096$, $B_V = 256$, $m_B = 0.9$, $N_{steps} = 3$ and $\gamma = 1.2$. Each feature transformer block uses one shared and two decision step-dependent fully-connected layer, ghost batch normalization and GLU blocks. Adam optimization is used with a learning rate of 0.01 (decayed 0.95 every 8k iterations with an exponential decay) for 600k iterations.

TabNet-M model uses $N_d = N_a = 64$, $\lambda_{sparse} = 0.0001$, $B = 4096$, $B_V = 128$, $m_B = 0.8$, $N_{steps} = 7$ and $\gamma = 1.5$. Each feature transformer block uses two shared and two decision step-dependent fully-connected layer, ghost batch normalization and GLU blocks. Adam optimization is used with a learning rate of 0.01 (decayed 0.95 every 8k iterations with an exponential decay) for 600k iterations.

The TabNet-L model uses $N_d = N_a = 128$, $\lambda_{sparse} = 0.0001$, $B = 4096$, $B_V = 128$, $m_B = 0.8$, $N_{steps} = 5$ and $\gamma = 1.5$. Each feature transformer block uses two shared and two decision step-dependent fully-connected layer, ghost batch normalization and GLU blocks. Adam optimization is used with a learning rate of 0.02 (decayed 0.9 every 8k iterations with an exponential decay) for 600k iterations.

## C.5  Higgs dataset

TabNet-S model uses $N_d = 24$, $N_a = 26$, $\lambda_{sparse} = 0.000001$, $B = 16384$, $B_V = 512$, $m_B = 0.6$, $N_{steps} = 5$ and $\gamma = 1.5$. Each feature transformer block uses two shared and two decision step-dependent fully-connected layer, ghost batch normalization and GLU blocks. Adam optimization is used with a learning rate of 0.02 (decayed 0.9 every 20k iterations with an exponential decay) for 870k iterations.

TabNet-M model uses $N_d = 96$, $N_a = 32$, $\lambda_{sparse} = 0.000001$, $B = 8192$, $B_V = 256$, $m_B = 0.9$, $N_{steps} = 8$ and $\gamma = 2.0$. Each feature transformer block uses two shared and two decision step-dependent fully-connected layer, ghost batch normalization and GLU blocks. Adam optimization is used with a learning rate of 0.025 (decayed 0.9 every 10k iterations with an exponential decay) for 370k iterations.

For gradient boosted trees, we use the implementation (Tensorflow, 2019). We choose the learning rate of 0.1 and optimize the maximum depth to 8, based on the performance. The Gradient boosted tree-S model uses 50 trees, the Gradient boosted tree-M model uses 300 trees and the Gradient boosted tree-L model uses 3000 trees.

## C.6  Mushroom edibility dataset

TabNet model uses $N_d = N_a = 8$, $\lambda_{sparse} = 0.001$, $B = 2048$, $B_V = 128$, $m_B = 0.9$, $N_{steps} = 3$ and $\gamma = 1.5$. Each feature transformer block uses two shared and two decision step-dependent

fully-connected layer, ghost batch normalization and GLU blocks. Adam optimization is used with a learning rate of 0.01 (decayed 0.8 every 400 iterations with an exponential decay) for 10k iterations.

### C.7 ADULT CENSUS INCOME DATASET

TabNet model uses $N_d = N_a = 16$, $\lambda_{sparse} = 0.0001$, $B = 4096$, $B_V = 128$, $m_B = 0.98$, $N_{steps} = 5$ and $\gamma = 1.5$. Each feature transformer block uses two shared and two decision step-dependent layer, ghost batch normalization and GLU blocks. Adam optimization is used with a learning rate of 0.02 (decayed 0.4 every 2.5k iterations with an exponential decay) for 7.7k iterations.

When only 40 labeled examples are used instead of the full dataset, based on re-optimization of hyperparameters on the validation set, we modify $B = 128$, $\lambda_{sparse} = 0.01$ and the learning rate of 0.005 (decayed 0.95 every 10 iterations with an exponential decay) for 100 iterations.

### C.8 ROSSMANN DATASET

TabNet model uses $N_d = N_a = 32$, $\lambda_{sparse} = 0.001$, $B = 4096$, $B_V = 512$, $m_B = 0.8$, $N_{steps} = 5$ and $\gamma = 1.2$. Each feature transformer block uses two shared and two decision step-dependent fully-connected layer, ghost batch normalization and GLU blocks. Adam optimization is used with a learning rate of 0.002 (decayed 0.95 every 2000 iterations with an exponential decay) for 15k iterations.

## D   GUIDELINES FOR HYPERPARAMETER SELECTION

We consider datasets ranging from $\sim$10K to $\sim$10M training points, with varying degrees of fitting difficulty. TabNet obtains high performance for all with a few general principles on hyperparameter selection:

- Most datasets yield the best results for $N_{steps} \in [3, 10]$. Typically, we observe that when there are more information-bearing features, the optimal value of $N_{steps}$ tends to be higher. On the other hand, increasing it beyond some value may adversely affect training dynamics as some paths in the network becomes deeper and there are more potentially-problematic ill-conditioned matrices. A very high value of $N_{steps}$ typically suffers from overfitting and yields poor generalization.

- Adjustment of the values of $N_d$ and $N_a$ is the most efficient way of obtaining a trade-off between performance and complexity. $N_d = N_a$ is a reasonable choice for most datasets. A very high value of $N_d$ and $N_a$ may suffer from overfitting and yield poor generalization.

- An optimal choice of $\gamma$ can have a major role on the overall performance. Typically a larger $N_{steps}$ value favors for a larger $\gamma$.

- A large batch size is beneficial for performance - if the memory constraints permit, as large as 1-10 % of the total training dataset size is suggested. The virtual batch size is typically much smaller than the batch size.

- Initially large learning rate is important, which should be gradually decayed until convergence.

When the model size is constrained, the hyperparameter search becomes more complicated. Because the optimal ways to increase the representation capacity may be chosen among different options, such as increasing the number of steps or the unit size. For example, increasing the number of units while slightly decreasing the step size can be a better way of optimal utilization of the limited capacity constrained by the size on the number of parameters.

## E   ABLATION STUDIES

In Table 10, we show the impact of various design and hyperparameter choices. For all cases, the number of iterations is optimized on the validation set.

Table 10: Ablation studies for the TabNet model for the forest cover type dataset.

| Ablation cases | Test accuracy % (difference) | Number of parameters |
|---|---|---|
| Base ($N_d = N_a = 64$, $\gamma = 1.5$, $N_{steps} = 5$, $\lambda_{sparse} = 0.0001$, feature transformer block composed of two shared and two decision step-dependent layers, $B = 16384$) | 96.99 (0) | 470k |
| Decreasing capacity via number of units (with $N_d = N_a = 32$) | 94.99 (-2.00) | 129k |
| Decreasing capacity via number of decision steps (with $N_{steps} = 3$) | 96.22 (-0.77) | 328k |
| Increasing capacity via number of decision steps (with $N_{steps} = 9$) | 95.48 (-1.51) | 755k |
| Decreasing capacity via all-shared feature transformer blocks | 96.74 (-0.25) | 143k |
| Increasing capacity via decision step-dependent feature transformer blocks | 96.76 (-0.23) | 703k |
| Replacing feature transformer block with a single shared layer | 95.32 (-1.67) | 35k |
| Replacing feature transformer block with a single shared layer, with ReLU instead of GLU | 93.92 (-3.07) | 27k |
| Replacing feature transformer block with two shared layers | 96.34 (-0.66) | 71k |
| Replacing feature transformer block with two shared layers and 1 decision step-dependent layer | 96.54 (-0.45) | 271k |
| Replacing feature transformer block with a single decision-step dependent layer | 94.71 (-0.28) | 105k |
| Replacing feature transformer block with a single decision-step dependent layer, with $N_d = N_a = 128$ | 96.24 (-0.75) | 208k |
| Replacing feature transformer block with a single decision-step dependent layer, with $N_d = N_a = 128$ and replacing GLU with ReLU | 95.67 (-1.32) | 139k |
| Replacing feature transformer block with a single decision-step dependent layer, with $N_d = N_a = 256$ and replacing GLU with ReLU | 96.41 (-0.58) | 278k |
| Reducing the impact of prior scale (with $\gamma = 3.0$) | 96.49 (-0.50) | 470k |
| Increasing the impact of prior scale (with $\gamma = 1.0$) | 96.67 (-0.32) | 470k |
| No sparsity regularization (with $\lambda_{sparse} = 0$) | 96.50 (-0.49) | 470k |
| High sparsity regularization (with $\lambda_{sparse} = 0.01$) | 93.87 (-3.12) | 470k |
| Small batch size ($B = 4096$) | 96.42 (-0.57) | 470k |

Obtaining high performance necessitates appropriately-adjusted model capacity based on the characteristics of the dataset. Decreasing the number of units $N_d$, $N_a$ or the number of decision steps $N_{steps}$ are efficient ways of gradually decreasing the capacity without significant degradation in performance. On the other hand, increasing these parameters beyond some value causes optimization issues and do not yield performance benefits.

Replacing the feature transformer block with a very simpler alternative, such as a single shared layer, can still give strong performance while yielding a very compact model architecture. This shows the importance of the inductive bias introduced with feature selection and sequential attention.

To push for the performance further, increasing the depth of the feature transformer is the effective approach. While increasing the depth, parameter sharing between feature transformer blocks across different decisions is an efficient way to decrease model size without significant degradation from performance. We indeed observe the benefit of partial parameter sharing, compared to fully decision step-dependent blocks or fully shared blocks. We observe the empirical benefit of GLU, compared to conventional nonlinearities like ReLU.

The strength of sparse feature selection depends on the two parameters we introduce: $\gamma$ and $\lambda_{sparse}$. We show that optimal choice of these two is important for performance. A $\gamma$ close to 1, or a high $\lambda_{sparse}$ may yield too tight constraints on the strength of sparsity and may hurt performance. On the other hand, there is still the benefit of a sufficient low $\gamma$ and sufficiently high $\lambda_{sparse}$, to aid learning of the model via a favorable inductive bias.

Lastly, given the fixed model architecture, we show the benefit of large-batch training, enabled by ghost batch normalization (Hoffer et al., 2017). The optimal batch size for TabNet seems considerably higher than the conventional batch sizes used for other data types, such as images or speech.

## F  MIXUP TRAINING

In (Zhang et al., 2017), mixup training was shown to be beneficial for tabular data learning, on small-scale datasets with simple neural network models comprising fully-connected layers. We experiment mixup training with TabNet and did not observe superior performance compared to standard softmax training. For Covertype dataset, the best mixup model (for mixup parameter $\alpha$=0.3) yields a test accuracy of 96.28%, roughly 0.7% lower than softmax training. For Higgs dataset, for the best TabNet-S model, the best mixup model (for mixup parameter $\alpha$=0.1) yields a test accuracy of 78.11%, roughly 0.1% lower than softmax training. We hypothesize that linearization of the inputs may cause significant shifts in the input distribution and thus adversely affect the feature selection blocks of TabNet.

## G  SEMI-SUPERVISED LEARNING

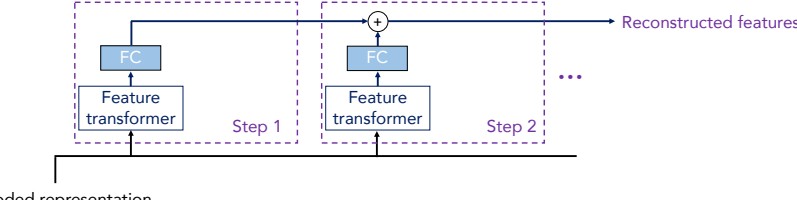

Figure 8: Decoder architecture to transform the encoded representation into reconstructed tabular data features. Each decision step is composed of a feature transformer block (see Fig. 3), and a fully-connected layer.

We explore the capability of TabNet in learning semantically-meaningful representations by integrating it into an autoencoder framework (Baldi, 2012). For this purpose, we propose a simple decoder architecture, shown in Fig. 8. The decoder is composed of a feature transformer block (as given in Fig. 3), followed by a fully-connected layer at each decision step. Different decision steps are summed to output the reconstructed features.

We propose an additive reconstruction loss (with a coefficient $\lambda_{unsup}$) between the input features $\mathbf{X}$ and the reconstructed features $\hat{\mathbf{X}}$. The reconstruction loss (computed over unlabeled data batch of size $B_U$) is in the form of L2 loss, normalized with the population standard deviation of the ground truth data, and scaled by $\mathbf{S}$:

$$L_{unsup}(\hat{\mathbf{X}}, \mathbf{X}) = \frac{1}{B_U \cdot D} \sum_{b=1}^{B_U} \sum_{j=1}^{D} \left| \frac{(\hat{\mathbf{X}}_{\mathbf{b,j}} - \mathbf{X}_{\mathbf{b,j}}) \cdot \mathbf{S}_{\mathbf{b,j}}}{\sqrt{\frac{1}{B_U} \sum_{b=1}^{B_U} (\mathbf{X}_{\mathbf{b,j}} - \frac{1}{B_U} \sum_{b=1}^{B_U} \mathbf{X}_{\mathbf{b,j}})^2}} \right|^2 \tag{7}$$

Normalization with the input value is observed to be crucial, as the tabular data features may have very different ranges. A straightforward approach in conventional reconstruction loss is scaling with a uniform mask, $\mathbf{S}_{\mathbf{b,j}} = 1/D$. As a more promising alternative, we propose that scaling should be based on feature importance values, such that the autoencoder should prioritize learning the representation for features that are the most important for decision making. We use the feature important mask $\mathbf{S}_{\mathbf{b,j}} = \mathbf{M}'_{\mathbf{agg-b,j}} / \sum_{j=1}^{D} \mathbf{M}'_{\mathbf{agg-b,j}}$ to promote learning for the most salient features. $\mathbf{M}'_{\mathbf{agg}}$ is inferred from the TabNet for the batch of unlabeled training samples, and fixed in the computation of loss to avoid the trivial solutions of fitting the easiest features.

For semi-supervised learning experiments, we consider the Adult Census Income dataset. We randomly choose 50 samples as the labeled set. We fix the TabNet model with the aforementioned hyperparameters. As the original learning hyperparameters overfit very quickly for 50 samples, we reoptimize the learning rate to 0.01 (decayed 0.9 every 100 iterations with an exponential decay) and trained for 800 iterations. For the autoencoder, we also fix the TabNet architecture, and optimize the decoder and learning hyperparameters. We use a decoder architecture with $N_d = N_a = 16$, $B = 128$, and $m_B = 0.98$. We use an unlabeled batch size of $B_U = 2048$. The model with uniform masking uses $\lambda_{unsup} = 0.2$, $\lambda_{sparse} = 0.005$, the number of decoder steps of $N_{steps} = 6$ and a learning rate of 0.005 (decayed 0.95 every 4k iterations with an exponential decay) and trained for 20.6k iterations. The model with feature importance mask uses $\lambda_{unsup} = 0.1$, $\lambda_{sparse} = 0.005$, the number of decoder steps of $N_{steps} = 4$ and a learning rate of 0.01 (decayed 0.9 every 10k iterations with an exponential decay) and trained for 63k iterations. Since feature importance masking focuses on reconstructing the most salient features, the learning capacity of the optimal decoder is lower.

Table 11: Results for semi-supervised learning for Adult Census Income, along with the supervised learning benchmarks. 50 samples with labels are randomly chosen from the training dataset. We reoptimize the learning hyperparameters on a separate validation set for a fair comparison.

| Dataset | Learning setting | Test accuracy (%) |
|---|---|---|
| 50 labeled | Fully-supervised | 76.8 |
| 50 labeled + 26015 unlabeled | Semi-supervised (autoencoder with uniform mask) | 78.9 |
| 50 labeled + 26015 unlabeled | Semi-supervised (autoencoder with feature importance mask) | 80.6 |
| 26065 labeled | Fully-supervised | 85.7 |

Table 11 shows the semi-supervised learning performance, along with the two supervised learning benchmarks: when trained without additional unlabeled data and when trained after labeling the entire dataset. We observe a significant boost in performance with the contributions from the unlabeled data, closing the gap towards the supervised learning baseline of the entire dataset. Focusing on the most important features in autoencoding helps improving the semi-supervised learning performance.

