# OpenReview forum: "TabNet: Attentive Interpretable Tabular Learning"
_ICLR.cc/2020/Conference — Reject_

### Official Review · AnonReviewer2 · 2019-10-23
**Official Blind Review #2**

**Rating:** 3

**Review:**

This paper proposes a new neural network for tabular data which uses sequential attention to perform instance-wise feature selection which can help learn superior decision rules, in addition to facilitating interpretability of the resulting model.  Overall, the paper is well-written and fairly easy to follow, and the idea appears conceptually well-grounded and of high practical value. However, I have concerns about the experiments.

As the proposed model is primarily empirically motivated (ie. there is no theoretical justification for why it should be better), it is imperative the authors conduct thorough experiments to convincingly demonstrate its utility.  However, the current experiments are too sparse to be fully convincing, given how easy it is nowadays to evaluate ML models across diverse collections of tabular datasets (eg. OpenML, UCI, CatBoost). While other papers on tabular-data models study many real datasets (eg. Klambauer et al), this paper only studies 4 real datasets (and does not even quantify the variability in results across multiple training/test splits).

Also, why did the authors not compare against xgboost/lightGBM/catboost, some of the most popular tabular models, in all datasets?

It seems different hyperparameter values were also used for each dataset, what was the search-space / hyperparameter optimization method used here? Or were hyperparameters for each dataset found through manual tuning.

Klambauer et al (2017). Self-Normalizing Neural Networks.
https://arxiv.org/abs/1706.02515

OpenML: https://www.openml.org

UCI: https://archive.ics.uci.edu/ml/datasets.php

CatBoost: https://catboost.ai/#benchmark

Other comments:

- Why don't the authors quantitatively compare the quality of the selected features from their model vs other feature selection methods for the Syn datasets?

- What is "softmax-training" in Appendix D? If the authors just mean standard training, I recommend they use this name instead.

- Since there are many components to this model, it would help clarify their purpose to the reader if the authors could provide one concrete example of the feedforward pass of a hypothetical instantiation of their model, say for a simple datapoint with just 2 features.

- the subscript i in h_i should match the i used in M[i], P[i] to avoid confusion

- P[0] (the base case) needs to be explained.

- a[i-1] should be more clearly defined, and the base case explained in more detail (I assume it is just the input features).


- I think it is a bit confusing to present the model parameters in terms of batch-size B, and think it would be better if you introduced them assuming batch-size = 1 (the generalization to batch-size B is obvious and should be familiar to readers as it's the same for all neural models).


**Experience Assessment:**

I have published in this field for several years.

**Review Assessment: Checking Correctness Of Derivations And Theory:**

I carefully checked the derivations and theory.

**Review Assessment: Checking Correctness Of Experiments:**

I carefully checked the experiments.

**Review Assessment: Thoroughness In Paper Reading:**

I read the paper thoroughly.

---

> ### Author Response · Authors · 2019-11-14
> **Response to Review #2**
>
> We really appreciate that you acknowledge our contributions, and finding our ideas “conceptually well-grounded and of high practical value”. We appreciate your specific suggestions below, which helped us to improve the quality of our paper.
>
> Q1: As the proposed model is primarily empirically motivated (ie. there is no theoretical justification for why it should be better), it is imperative the authors conduct thorough experiments to convincingly demonstrate its utility.  However, the current experiments are too sparse to be fully convincing, given how easy it is nowadays to evaluate ML models across diverse collections of tabular datasets (eg. OpenML, UCI, CatBoost). While other papers on tabular-data models study many real datasets (eg. Klambauer et al), this paper only studies 4 real datasets (and does not even quantify the variability in results across multiple training/test splits).
>
> A1: Thanks for this suggestion. We have run experiments on one more real-world dataset (Rossmann Store Sales) and added the results to the paper (see Appendix B). We think this case is specifically important because it has time features, unlike other datasets we considered. As requested we also run additional experiments on the CatBoost benchmarks (see below). Unfortunately most other UCI/OpenML datasets (particularly the 121 ones in Klambauer et al.) are too small in size to represent high-value real-world use cases where advanced machine learning methods can make a significant difference and it can be shown in statistically-significant way.  If you have any other suggestions we’d also be happy to add them to the paper later.
>
> Q2: Also, why did the authors not compare against xgboost/lightGBM/catboost, some of the most popular tabular models, in all datasets?
>
> A2: Following your suggestion, we have obtained results with these baselines and include them. For the poker hand induction example in Table 2, XGBoost achieves 71.1%, CatBoost achieves 66.6% and LightGBM achieves 70.0% accuracy (after extensive tuning), which are significantly worse than TabNet, which achieves 99.3%. For the Sarcos robotics arm inverse dynamics example in Table 3,  XGBoost achieves 0.21, CatBoost achieves 0.20 and LightGBM achieves 0.17 MSE, which are again significantly worse than the best TabNet, which achieves 0.14 MSE.
>
> Q3: It seems different hyperparameter values were also used for each dataset, what was the search-space / hyperparameter optimization method used here? Or were hyperparameters for each dataset found through manual tuning.
>
> A3: We start with a pre-defined hyperparameter value space. For example, N_d and N_a should be in {8, 16, 24, 32, 64, 128}. If the model size is not under the desired cutoff (e.g. for Table 5 comparisons), we decrease the value to satisfy the model size constraint. We do not tune the hyperparameters very extensively. We expect advanced and more extensive hyperparameter tuning to further improve the results. We have added more clarifications to Appendix.
>
> Other comments:
>
> Q4: Why don't the authors quantitatively compare the quality of the selected features from their model vs other feature selection methods for the Syn datasets?
>
> A4: We think that the overall performance is a more direct indication of the effective feature selection for these synthetic datasets. If we look at a simple metric like the ratio of the average mask coefficients for the features that are known to be salient to the features that are known to be irrelevant, we can observe that it is indeed very high for TabNet in most cases, e.g. 50.1 for the Syn2 dataset. Fig. 4 and 5 further build qualitative insights on the efficiency of feature selection (but on a larger scale version of Syn datasets).
>
> Q5: What is "softmax-training" in Appendix D? If the authors just mean standard training, I recommend they use this name instead.
>
> A5: Thanks for the suggestion, we have changed as indicated. By softmax-training we meant standard classification training with softmax cross entropy loss, as in conventional deep learning practices.
>
> Q6: Since there are many components to this model, it would help clarify their purpose to the reader if the authors could provide one concrete example of the feedforward pass of a hypothetical instantiation of their model, say for a simple datapoint with just 2 features.
>
> A6: Following your suggestion, we have added a diagram for the simplified feedforward pass to Appendix A.
>
> Note that we also have Fig. 4 which shows a toy synthetic dataset with 11 features where most are redundant. The masks show that mostly one feature is used at each of the 4 decision steps and the model eventually aggregates the information.

---

> > ### Author Response · Authors · 2019-11-14
> > **Response to Review #2 - continued**
> >
> > Q7: the subscript i in h_i should match the i used in M[i], P[i] to avoid confusion
> >
> > A7: We use subscripts for the functions and the square brackets for the variables for notational clarity as h[i](a[i-1]) would be more confusing.
> >
> > Q8: P[0] (the base case) needs to be explained.
> >
> > A8: Thanks for finding this important point that we ignored to include - P[0] is initialized as all ones, we have added a note.
> >
> > Q9: a[i-1] should be more clearly defined, and the base case explained in more detail (I assume it is just the input features).
> >
> > A9: Eq. (4) formulates how a[i] is obtained (we have switched the order of left-hand size and right-hand side for clarity), as the output of the feature transformer.
> >
> > Q10: I think it is a bit confusing to present the model parameters in terms of batch-size B, and think it would be better if you introduced them assuming batch-size = 1 (the generalization to batch-size B is obvious and should be familiar to readers as it's the same for all neural models).
> >
> > A10: Thanks for the suggestion. We aimed to be more explicit about the matrix sizes and the definition of some of the loss functions that are averaged across the batch.
> >
> > We hope that we have fully addressed your questions and concerns. Please let us know if you have further comments.

---

> > > ### Comment · AnonReviewer2 · 2019-11-15
> > > **Comments on revision**
> > >
> > > I applaud the authors for greatly improving their paper via the revision.
> > >
> > > That said, I still believe that the authors should have experimented with many more tabular datasets to truly convince the reader of the proposed method's utility.  There exist many well known (and already nicely processed in CSV format) tabular datasets which have sufficiently large sample-size to statistically distinguish different methods (and are not computationally expensive to train on).  It should be very easy for the authors to run a standard feedforward network and tree ensemble like RF/XGBoost/lightGBM on many such datasets and simply report the results in a big table.  The fact that so many of the comparison numbers are coming from different papers is concerning to me as well, again it seems easy to run at least 1-2 baselines yourself on every dataset given how standardized most tabular datasets already are.

---

> > > > ### Author Response · Authors · 2019-11-15
> > > > **Thanks for appreciating the improvements & Further Results**
> > > >
> > > > Thanks for appreciating the improvements from rebuttal!
> > > >
> > > > Following your suggestion, we have added results on two more datasets (now 6 new datasets during rebuttal period) to Appendix B and will continue adding more results to the final revision of the paper (we only had 7 daylight hours since this comment so we could not yet run even more). We now have results for 6 synthetic datasets and 11 real-world datasets (Forest cover type, Poker, Sarcos, Higgs, Mushroom, Rossmann, Appetency, Churn, Upselling, Census Income, Loan) and we compare TabNet to more than 10 tabular data learning methods to benchmark its performance. None of the tabular data learning papers in literature consider as many benchmarking methods. For example, we consider advanced deep learning methods like sparse evolutionary training, while the papers in ensemble decision tree learning community rarely compare their methods to any neural network-based solution.
> > > >
> > > > Regarding using comparison numbers from different papers, we did this to make sure we represent the competing methods in their best light as the authors who wrote the papers are more knowledgeable about their method and can better optimize the parameters.  However for almost all methods we also ran them ourselves to verify the numbers, and for any that we did not yet we will do so.  As far as we can tell, using numbers from other papers seems to be best practice in most ML papers, so we followed that practice.
> > > >
> > > > Overall, we really hope that you appreciate our contributions in this paper and our additional efforts to improve it during the rebuttal.  In this rebuttal period we have already added results on 6 additional datasets, and if necessary can add even more post rebuttal.  Thanks again for your helpful suggestions to improve the paper.

---

### Official Review · AnonReviewer1 · 2019-10-24
**Official Blind Review #1**

**Rating:** 3

**Review:**

This paper proposes a novel deep machine learning model called TabNet for learning from tabular model. The system builds on sequential feature acquisition that is aggregated to perform decisions. The performance of TabNet is evaluated in a set of experiments.

The paper proposes a network architecture which seems to achieve good performance on a set of experiments. However, I am missing a motivation for parts of the model. A broader ablation study in which certain components of the model are replaced by simpler network structures could be instructive; the current ablation study only investigates hyper-parameters and a few design choices -- but what happens for example if a feature transformer is simply replaced by fully connected layer instead of the used architecture, i.e. how general is the structure? Regarding experiments I am also missing details and a broader comparison with other methods.

Some more questions/comments:
* What is the training criterion used?
* In Table 1, what precisely is AUC? Area under "accuracy over selected features"? If so, what does the number "No sel." mean?
* Why is "No sel." performing relatively bad? Overfitting?
* What about stronger feature selection baselines, like those based on Shapley values?
* What do the errors in table 1 mean? Which numbers are marked in bold? This seems misleading when looking at the "error bars".
* What are the std errors for all other tables?
* What is the "ratio" in Figure 6a? Why does the figure not show INVASE?
* There are other instance-wise feature selection methods, e.g. Chao Ma et al.'s "EDDI: Efficient Dynamic Discovery of High-Value Information with Partial VAE" or . How does the proposed method compare against those?
* For making a claim about tabular data in general, more datasets should be considered.

Minor comments:
* Figure 1: It would support the reader to explain the blocks in the figure caption.

**Experience Assessment:**

I have published one or two papers in this area.

**Review Assessment: Checking Correctness Of Derivations And Theory:**

N/A

**Review Assessment: Checking Correctness Of Experiments:**

I assessed the sensibility of the experiments.

**Review Assessment: Thoroughness In Paper Reading:**

I made a quick assessment of this paper.

---

> ### Author Response · Authors · 2019-11-14
> **Response to Review #1**
>
> Thanks for your valuable comments overall and specific suggestions below, which helped us to clarify our contributions and improve the quality of our paper.  Please see below for answers to questions as well as the requested additional experiments.
>
> Q1: The paper proposes a network architecture which seems to achieve good performance on a set of experiments. However, I am missing a motivation for parts of the model. A broader ablation study in which certain components of the model are replaced by simpler network structures could be instructive; the current ablation study only investigates hyper-parameters and a few design choices -- but what happens for example if a feature transformer is simply replaced by fully connected layer instead of the used architecture, i.e. how general is the structure? Regarding experiments I am also missing details and a broader comparison with other methods.
>
> A1: Following your suggestion, we have expanded the list of design choices in the ablation studies section (please see our updated Table 7) to show the generality of the core idea of the TabNet architecture, based on applying sequential decision-making with attentive feature selection.
>
> Details of some design choices beyond the core idea are also somewhat important to push the performance as high as possible. For example, Table 7 shows that having the feature transformer blocks partially shared yields better performance compared to having them entirely shared or having them entirely decision step dependent. Regarding replacing the feature transformer with simpler networks, we observe some performance degradation (~1.5% for forest cover type, see the updated Table 7) compared to the proposed best model, while still outperforming most other comparison benchmarks significantly.
>
> We have provided clarifications to the missing details and comparisons below.
>
> Some more questions/comments:
>
> Q2: What is the training criterion used?
>
> A2: We have clarified the training criterion - we use standard classification (softmax cross entropy) and regression (mean squared error) loss functions and we train until convergence, similar to other conventional deep learning methods.
>
> Q3: In Table 1, what precisely is AUC? Area under "accuracy over selected features"? If so, what does the number "No sel." mean?
>
> A3: AUC is the area under the receiving operating characteristic curve, for the predicted output. No sel. refers to the case we use all the 11 features to train the predictor model. We have clarified these in the updated version.
>
> Q4: Why is "No sel." performing relatively bad? Overfitting?
>
> A4: The synthetic datasets are constructed in such a way that only a subset of the features affect the ground truth output. If no feature selection is performed, the model would overfit while relating the predicted output to irrelevant features.
>
> Q5: What about stronger feature selection baselines, like those based on Shapley values?
>
> A5: We compare our method to two very strong feature selection baselines that are recently published - L2X [1] and INVASE [2]. Indeed, in [2], L2X is compared to feature selection based on Shapley values and significant outperformance of L2X was demonstrated. As given in Table 1, TabNet clearly outperforms L2X.
>
> [1] “Learning to explain: An information-theoretic perspective on model interpretation”, by J. Chen et al.
> [2] “INVASE: Instance-wise variable selection using neural networks”, by J. Yoon et al.
>
> Q6: What do the errors in table 1 mean? Which numbers are marked in bold? This seems misleading when looking at the "error bars".
>
> A6: The numbers correspond to mean and std values of AUC of different runs, similar setting to INVASE paper [2]. Bold numbers are the best method for each dataset. We have added clarification.
>
> Q7: What are the std errors for all other tables?
>
> A7: Other datasets are very large scale so randomness is minor. We did not include multiple runs due to compute constraints.
>
> Q8: What is the "ratio" in Figure 6a? Why does the figure not show INVASE?
>
> A8: Ratio means the ratio of the feature importance of “Odor” feature to all the features (e.g. if the ratio is 100%, it becomes the only important feature), we have clarified this. We will also include INVASE.

---

> > ### Author Response · Authors · 2019-11-14
> > **Response to Review #1 - continued**
> >
> >
> > Q9: There are other instance-wise feature selection methods, e.g. Chao Ma et al.'s "EDDI: Efficient Dynamic Discovery of High-Value Information with Partial VAE" or . How does the proposed method compare against those?
> >
> > A9: We have cited this work and explain how it compares. Essentially, EDDI is proposed to dynamically decide which piece of information to acquire next - it is sequential and it does not require fully observed dataset. Thus, the setting is different than our experiments and would not constitute a meaningful comparison. (Their Authors explain here: https://openreview.net/forum?id=HJl0jiRqtX in detail how their setting is different from conventional feature selection.)
> >
> > Q10: For making a claim about tabular data in general, more datasets should be considered.
> >
> > A10: We focus on a wide range of problems that are commonly considered in other academic papers. We have added results on an additional real-world dataset (see Appendix B) and carefully revised our claims.
> >
> > Minor comments:
> > Q11: Figure 1: It would support the reader to explain the blocks in the figure caption.
> >
> > A11: Thanks for the suggestion, we have added a few more sentences to expand the explanations for the blocks in the figure caption.
> >
> > We hope that we have fully addressed your questions and concerns. Please let us know if you have further comments.

---

### Official Review · AnonReviewer3 · 2019-10-26
**Official Blind Review #3**

**Rating:** 6

**Review:**

In this paper, the authors propose a novel architecture with sequential attention modules for tabular learning. An attention module is trained to select some elements from the (normalized) input feature, and a feature transformer takes the selected features for overall feature embedding. The model is evaluated on multiple tasks, and the proposed method outperforms prior approaches in the paper.

I am in favor of this paper as it proposes an interpretable method for feature extraction in tabular learning. The learned overall feature embedding is shown to be effective in multiple tasks. It seems to me that the proposed method may have high generalization capability because the method is trained to select sparse feature attributes for decision. The authors may study the generalization performance further.

The experiment section may not be very reliable. The numbers indicated with * are copied from other papers, but these numbers may not be compatible if there is any discrepancy in the experiment setup. It would be better if the authors can run the baselines on the datasets.

I am a little disappointed that no study on the number of steps needed for the model. It seems non-trivial to me. In TabNet-L, the step size is 5 but in TabNet-M, the step size is 7 (even smaller?). How should we choose the right step size?

**Experience Assessment:**

I have read many papers in this area.

**Review Assessment: Checking Correctness Of Derivations And Theory:**

I assessed the sensibility of the derivations and theory.

**Review Assessment: Checking Correctness Of Experiments:**

I assessed the sensibility of the experiments.

**Review Assessment: Thoroughness In Paper Reading:**

I read the paper at least twice and used my best judgement in assessing the paper.

---

> ### Author Response · Authors · 2019-11-14
> **Response to Review #3**
>
> Thanks for your valuable thoughts and accurate interpretation of our high level ideas on how high generalization capability can be obtained in conjunction with interpretability.
>
> Q1: The experiment section may not be very reliable. The numbers indicated with * are copied from other papers, but these numbers may be compatible if there is any discrepancy in the experiment setup. It would be better if the authors can run the baselines on the datasets.
>
> A1: We agree that reliability of experiments is very important and it is something we constantly had in mind when designing experiments.  While copying results from other papers, we made sure to use the same experimental setup such as the exact same split of training, validation and test sets.  We also ran their codebases whenever they were available.
>
> Q2: I am a little disappointed that no study on the number of steps needed for the model. It seems non-trivial to me. In TabNet-L, the step size is 5 but in TabNet-M, the step size is 7 (even smaller?). How should we choose the right step size?
>
> A2: The number of steps is an important hyperparameter and its optimal value may depend on the dataset. We observe that when there are more information-bearing features, the optimal value of the number of steps tends to be higher (e.g. when we compare Syn1 and Syn5 datasets). On the other hand, increasing it beyond some value may adversely affect training dynamics as some paths in the network becomes deeper and there are more potentially-problematic ill-conditioned matrices. Our ablation study in Table 7 shows that for forest cover dataset, increasing or decreasing the optimal number of steps may reduce performance. We have expanded our discussions in Appendix C where we provide guidance on the selection of hyperparameters.
>
> When the model size is constrained, the hyperparameter search becomes even more complicated. Because the optimal ways to increase the representation capacity may be chosen among different options, such as increasing the number of steps or the unit size. As you indicated, in some examples, increasing the number of units while slightly decreasing the step size can be a better way of optimal utilization of the limited capacity constrained by the number of parameters. We have expanded the explanations and described the different tradeoffs in a bit more detail.
>
> We hope that we have provided further insights on your questions. Please let us know if you have further comments.

---

### Public Comment · ~Even_Oldridge1 · 2019-09-30
**Interesting architecture, but the performance evaluation seems weak.**

The architecture and idea are interesting.  I'm not sure about novelty; it sounds very close to AutoInt which uses self attention to aggregate features into higher order representations.  Certainly the AutoInt paper should be cited and contrasted against to demonstrate novelty.  https://arxiv.org/abs/1810.11921

Performance evaluation is weak IMO.  The authors talk about comparing against xgboost and lgbm but only do so on a single dataset.  Each comparison is against a different set of algorithms and the paper would be much more solid if the comparisons were consistent.  The synthetic datasets used for the first part of evaluation are in my opinion not relevant in the real world (10K samples) and I'm not sure they provide any value.  The other four datasets aren't common in my reading of other tabular papers and the comparisons as mentioned are each against different methods.

As a starting point I'd recommend evaluating against the datasets posted in the catboost benchmarks: https://github.com/catboost/benchmarks/tree/master/quality_benchmarks

They're flawed in that logloss isn't the most important metric, and can be misleading relative to ROC and accuracy, but it would still be better than the evaluation that was performed.

---

> ### Author Response · Authors · 2019-10-08
> **Fundamental differences with AutoInt and more on performance results**
>
> Thanks for the valuable feedback!
>
> We will cite the AutoInt paper and explain the clear differences. Although both TabNet and AutoInt utilize attention in their architecture for selection of the input features, there is a very fundamental distinction in the use of attention: TabNet uses sequential top-down attention by generating the queries at each decision step vs. AutoInt uses conventional self-attention by generating queries locally for each feature. With sequential top-down attention, jointly-processed information from the previous decision step directly determines what features to focus on at the current decision step. On the other hand, AutoInt uses self-attention to explicitly model pairwise feature interactions. TabNet yields D-dimensional feature importance masks at each decision step, whereas AutoInt yields DxD-dimensional attention activation maps at each head (where D is the number of features). Thus, there are differences in terms of representation capacity, model size and interpretability. In addition to attention for feature selection, TabNet architecture also introduces an efficient mechanism to control the sparsity of the selected features, in order to allocate the learning capacity to salient features for compact modeling.
>
> Regarding our evaluations, we specifically focused on the results that are published in other papers, thus different datasets include different set of comparisons. Analyzing most of the published tabular data learning papers in the past few years, Higgs and Forest cover type datasets seem the most common datasets in academic literature (without consistency in validation/test splits or performance metrics though). Tabular datasets are very diverse in real-life applications and unfortunately there aren't a few representative public datasets that the academia can focus on - even the CatBoost benchmarks do not seem to be systematically used in other papers (indeed we have yet to find a single paper that compares their results to all CatBoost benchmarks).
>
> The synthetic datasets are constructed in such a way that only a subset of the features determine the output. Thus, they can be used for evaluation of the efficacy of feature selection (by comparing to the cases when all features vs. only the salient features are used), as in other published papers.
>
> On the 3 KDD datasets from the CatBoost benchmarks, we have tried TabNet with the same experimental settings and using the same limited number of hyperparameter trials. Our results seem very close to the top results of CatBoost: in terms of zero-one loss, on Appetency TabNet yields 0.018 vs. the best CatBoost yields 0.018; on Churn TabNet yields 0.073 vs. the best CatBoost yields 0.072; and on Upselling TabNet yields 0.050 vs. the best CatBoost achieves 0.049. We will include CatBoost benchmark comparisons in our paper as well.

---

### Public Comment · ~John_Ryan3 · 2019-10-19
**More methods should be compared**

Some competitive CTR prediction methods should be involved, like DeepFM and AutoInt.

So, this paper only conducts feature selection? No feature interaction is considered? I notice there’s FC layer, the interaction is done by this? I think this harms the interpretability.

---

> ### Author Response · Authors · 2019-10-25
> **Further explanations on feature selection and AutoInt comparison**
>
> Thanks for your suggestion. We have experimented AutoInt architecture (which already shows slight outperformance of DeepFM) with the same inputs on the forest cover type dataset (see Table 2). After extensive hyperparameter search, the best AutoInt model achieves 90.24% accuracy - significantly lower than TabNet.
>
> TabNet employs instance-wise feature selection at each decision step, and then processes the selected inputs jointly with the feature transformer block (please refer to Fig. 1). Thus, the interaction between different features would be considered if the model infers that it is the appropriate processing to apply. Consider the example depicted in Fig. 5 - at each decision step, you can observe that multiple features are selected (and sometimes it is exactly two features), and then processed jointly.  The joint processing of multiple features is done with our transformer block, which is deep and nonlinear.
>
> The last FC layer is linear indeed, merely for the purpose of dimensionality matching. The interpretability comes from the feature selection, and it is approximate as our feature processing is nonlinear (please see Section 4.2). There are options to make the architecture more interpretable (e.g. by having linear feature processing), but it would come at a cost of significantly decreased performance. Overall, we believe that TabNet design makes a significant leap forward towards jointly achieving interpretable and high-performance tabular data learning.

---

### Decision · Program_Chairs · 2019-12-19

**Decision:**

Reject

**Comment:**

This paper constitutes interesting progress on an important topic; the reviewers identify certain improvements and directions for future work, and I urge the authors to continue to develop refinements and extensions.